# Effects of Mindfulness-Based Interventions (MBIs) in Patients with Early-Stage Alzheimer’s Disease: A Pilot Study

**DOI:** 10.3390/brainsci13030484

**Published:** 2023-03-13

**Authors:** M. V. Giulietti, R. Spatuzzi, P. Fabbietti, A. Vespa

**Affiliations:** 1Scientific and Technological Area, Department of Neurology, INRCA-IRCCS National Institute of Health and Science on Aging, 60124 Ancona, Italy; 2Department of Mental Health, ASP Basilicata, 85100 Potenza, Italy; roberta.spatuzzi@yahoo.com; 3Biostatistical Center, INRCA-IRCCS National Institute of Science and Health on Aging, 60124 Ancona, Italy; p.fabbietti@inrca.it

**Keywords:** mindfulness, cognitive status, neuropsychiatric symptoms, depression, quality of life, Alzheimer’s disease

## Abstract

Bachground In this study, we hypothesize that mindfulness-based interventions (MBIs) may improve well-being and the related outcomes in Alzheimer’s dementia patients (AD-P) at an early stage. MBIs consist of the practice of consciously observing the psychic contents in the present moment (thoughts, sensations, feelings, and other events). This attention allows one to become aware of the psychic contents and integrate them, thus favoring the quality of life and an increase in the mood of practitioners. Methods The randomized controlled study enrolled 22 AD-P at an early stage (age ≥ 60 years) treated with MBIs and 22 patients without treatment (six months of MBI training). Tests (T0–T1 six months): Mini-Mental State Examination (MMPI); Spiritual Well-Being (SWB); Beck Depression Inventory (BDI); SF36. Test-Caregiver: Everyday Cognition scales (ECOG). Results AD-P with mindfulness: Improvement of ECOG (*p* = 0.026), quality of life (*p* < 0.001), spiritual well-being (*p* < 0.001); decrease in depression BDI (*p* < 0.001). The MMSE remains unchanged. The control group of untreated patients showed a significant worsening in all these dimensions. Conclusions MBI training is effective in increasing quality of life and preventing worsening in patients with early-stage Alzheimer’s dementia.

## 1. Introduction

With the increase in the elderly population, a significant increase in the incidence of cognitive impairment is expected [1]. Alzheimer’s disease (AD) is the leading cause of cognitive impairment and dementia in older individuals (age ≥ 65 years) worldwide, accounting for 60–80% of dementia cases. The symptoms of patients with AD are progressive deficits in cognitive, emotional, and physical functions (e.g., loss of memory and other cognitive abilities) that are severe enough to cause functional disability and a loss of independence [2]. Furthermore, it also involves serious behavioral and psychological alterations (i.e., depression, agitation, wandering, aggression, depression) [3]. Risk factors for AD include age, gender, genetics, and environmental factors [4,5,6]. The patients have difficulty in the controlled expression of feeling: they overreact to things, show sudden changes in mood, or feel irritable, which is accompanied by dysfunctional coping strategies, such as wandering, withdrawal, avoidance, and aggressive actions [7]. They also have difficulties managing pain.

AD Patients (AD-P) present generalized anxiety states (GAD) with persistent and excessive nagging about a number of different things [8]. Such anxiety manifests itself through restlessness, fatigue, concentration problems, irritability, muscle tension, and sleep disturbances during the course of dementia in the mild stages, whereas in the severe stage of dementia, anxiety decreases [9,10]. Mild to moderate AD patients with generalized anxiety show significantly higher scores of depression, irritability, overt aggression, mania, and pathological crying than AD patients without it but with similar age and cognitive impairment, and symptoms such as restlessness, muscle tension, fears, insomnia, and physical ailments are the most important symptoms [7]. So, anxiety is correlated with a low quality of life.

There are many factors associated with a good or poor quality of life (QoL). The caregiver relationship, religiosity, social engagement, and the ability to engage in activities of daily living (ADL) were positively associated with a good QoL, while neuropsychiatric symptoms were associated with a poorer QoL.

Some authors have suggested that lifestyle changes (environmental factors—behavioral changes, such as being physically and cognitively active, being socially engaged, and eating a heart- and brain-healthy diet) could prevent 30% of AD cases [4,5,6].

The degeneration of the personality of the patient with AD creates a considerable burden for the family and the caregiver, with consequences for his/her psycho-physical health [11]. AD patients become dependent on others for their daily needs, and this dependence increases over time.

Traditionally, the main approach to depression and neuropsychiatric symptoms in AD patients is pharmacological treatment, although there are insufficient data to justify this and research needs to improve. Many authors highlight that pharmacological therapy provided little effect and recommend non-pharmacological therapies [12,13,14]. However, there is some consensus suggesting that the main approach to AD should start with non-pharmacological treatment [13].

Unfortunately, even cognitive stimulation shows limits in promoting significant improvements in cognition and the reduction of neuropsychiatric symptoms with limited and/or no transfer effects in daily activities, in cognition, and in the reduction of neuropsychiatric symptoms [15,16,17]. The authors hypothesize that this is presumably due to an inability to improve moment-to-moment attention control while simultaneously reducing distractions from internal and external interruptions [18].

Some non-pharmacological approaches, such as cognitive behavioral therapy and music therapy have been proposed as alternatives for elderly patients with anxiety comorbidity with mild dementia or immediately after the diagnosis of dementia. Other authors highlight that exercise, aromatherapy, and massage were effective, in that order, as treatments to reduce anxiety in patients with dementia [19].

Promising interventions are mindfulness-based interventions (MBIs), which, as opposed to mere cognitive training, improve moment-to-moment attention control while simultaneously [20,21,22,23,24,25] reducing distractions due to internal and external interruptions. The other benefits of these meditation or mindfulness practices are a reduction in stress and an improvement in mood, quality of life, and socio-emotional and cognitive balance in general [23,26]. In fact, MBIs constitute primary prevention interventions for health in general and can be used as secondary prevention in the field of physical and mental health, even in neurodegenerative diseases such as AD.

Furthermore, stress, depression, anxiety, and neuroticism affect sleep, cognition, and mental health and well-being in aging populations and are associated with an increased risk of AD and further degeneration [3].

In fact, some studies show that the practice of MBI, favoring the controlled attention component and, therefore, the cognitive control of active memory (working memory), at the present moment, without judging the emotional or thought contents, could help reduce these adverse factors [27,28,29,30], reducing response interference [31,32,33,34]. Given these premises, we implemented a 6-month MBI treatment program for patients with early-stage Alzheimer’s dementia. Mindfulness programs usually last 2 months but we considered such an intervention insufficient for patients with cognitive impairment [31]. If our results are positive, it will be possible to propose the intervention with mindfulness as a good clinical practice for patients with AD.

## 2. Materials and Methods

### 2.1. Design

Dyads of caregivers and patients with Alzheimer’s dementia (early-stage) were consecutively interviewed.

### 2.2. Participants

One hundred seniors affected by Alzheimer’s disease were approached in the clinic by the physician and asked to participate in the study. All the participants signed a consensus form regarding the study protocol after a detailed explanation by the physician at the neurology clinic. Only one hundred and five individuals decided to participate and fill out and sign the consent form. They were diagnosed with Alzheimer’s disease at an early stage and were enrolled in the dementia disease assessment clinic of neurology of the IRCCS-INRCA Hospital in Ancona. Most of them were living with family, and all were in touch daily with family members. The approval number of the Bioethical Advisory Committee of IRCCS-INRCA, Ancona, Italy is IRB n.19009.

Fifteen patients did not answer all the questions in the questionnaires or meet the inclusion criteria. It was, therefore, decided not to consider them for the analysis.

### 2.3. Inclusion Criteria

Age > 70; diagnosis of Alzheimer’s disease (early-stage) according to NINDS-ADRDA criteria; MMSE score between 18 and 27; availability during testing and intervention; the presence of caregiver; and willingness to sign informed consent [35].

### 2.4. Exclusion Criteria

Subjects with severe medical conditions: neurological disorders such as Parkinson’s disease; other dementias; primary brain neoplasia; untreated epilepsy; subjects with sensorimotor deficits; and the presence of sensory impairments that interfere with treatment subjects with severe psychopathological comorbidities.

### 2.5. Randomization

The AD Patients (n = 80) were randomly allocated to the groups of treatment-MBIs practice (40 AD-P) or the control groups without treatment (40 AD-P) using the randomization list.

The treatment of the other subjects included in the study had to be stopped due to the COVID-19 lockdown.

So, 22 patients were treated with mindfulness in the group and 22 AD-P did not receive treatment.

### 2.6. Intervention Structure

The participants took part in one 1-h session each week taught by a single psychotherapist with specific training in mindfulness and ten years of experience in meditation practice.

The patients attended the following structured training in weekly meetings for six months:

First month: learning stress management exercises: Jacobson relaxation technique (for 1 and 1/2 min 3 times a week).

The expected effects are the following: relaxation of the entire psycho-physical organism and general calmness. 

After the first month of relaxation training, the group session involves learning the meditative practice of mindfulness (mindfulness-based interventions (MBIs)). The patient starts exercising with MBIs for 15–20 min three times a week (two times at home and once in the therapeutic setting), continuing the training with relaxation.

The expected effects of the MBI practice are the following: increased self-awareness and effortless attention to the present moment; the reduction of ruminations; psychological well-being; the reduction of psychological symptoms; the reduction of stress; the reduction of related neuropsychiatric symptoms; the overcoming of depression; and emotional-cognitive rebalancing.

### 2.7. Interviews (T0–T1 Six Months)

The interview (1 h) was taught by a psychotherapist trained in the specific fields of Humanistic Existential Psychotherapies, Mindfulness Training, Neuropsychology, and Palliative Care with a single patient (belonging to the treated and non-treatment groups) at T0 and T1.

The purpose of the interview was to identify the difficulties and problems of the individual patient (of treated groups) to be faced in group work.

A second meeting was aimed at administering the tests.

The control group was left without any intervention in order not to include other treatments that could have been confounding factors of the results. These patients were assigned to cognitive training groups as standard treatment, delivered by neuropsychologists at our Alzheimer’s Center-INRCA at the end of the study (after six months).

### 2.8. Tests

All the patients were asked to complete the following tests:(a)**The Mini-Mental State Examination (MMSE)** [36], which assesses the presence or absence of dementia using 30 questions that are descriptive of the following dimensions: orientation, concentration, attention, verbal memory, naming, and visuospatial skills. The MMSE is easy to apply, requires no technical or expensive equipment, and can be completed within 5–10 min. The maximum score is 30, which indicates no cognitive impairment. The cutoff score is 23–24, and most non-demented older adults rarely score below 24. It is a questionnaire used extensively in clinical and research settings.(b)**The SF-36** [37,38], which is a popular tool for assessing HRQoL and has been used in many physical health conditions and healthcare settings. It defines HRQoL as the extent to which physical health impacts an individual’s functional ability and perceived well-being in mental, social, and physical aspects of life and assesses various dimensions of quality of life by 36 items. The SF-36 items represent multiple operational definitions of health, including function and dysfunction, distress and well-being, objective reports and subjective ratings, and both favorable and unfavorable self-evaluations of general health status. It encompasses eight domains: physical functioning (PF), health role limitations (HR), role limitation emotional problems (RE), energy fatigue (E), emotional well-being (EWB), social functioning (SF), pain (P), and general health (GH). The score ranges from 0 (worst overall health) to 100 (best overall health) for each of the eight domains.(c)**The Beck Depression Inventory (BDI)** [39], which describes the severity of depression using 21 questions, and the user is asked to score these statements in relation to how they have felt in the past 2 weeks, including today. The 4-point scale ranges from 0 to 3. The BDI-II is evaluated by adding the highest scores for each of the 21 items. It has reliability and validity for use in clinical trials. Therefore, the total score can range from 0 to 63. A score of 0–9 indicates not depressed, 10–15 indicates mildly depressed, 16–24 indicates moderately depressed, and a score of 25 or more signifies severely depressed.(d)**Spiritual Well-Being Index FACIT-SP (SWB)** [40], which describes spiritual well-being and has the reliability and validity to be used in clinical trials with patients diagnosed and treated for serious diseases. The responses are rated on a 5-point Likert scale (0 ¼ not at all; 4 ¼ very much). The subscales are as follows: meaning/peace (items 1–8) and the faith subscale (comfort and strength in one’s spiritual beliefs) (items 9–12). The subscale scores are the sum of the items (meaning/peace out of 32 and faith out of 16).

All the caregivers were asked to fill out the following tests:
**Social schedule**: data on sex, age, marital status, educational level, profession, and “living with”.**The Everyday Cognition (ECOG)** activities of daily living [41]. Assesses a patient’s level of functioning in terms of their ability to care for themself, their daily activity, and their physical ability (walking, working, etc.). The choice is among 5 items (a patient who is able to carry out all normal daily activities without effort (0) to a completely disabled patient (4)).**Neuropsychiatric Inventory (NPI)** [42]. Evaluates neuropsychiatric symptoms in dementia. The NPI examines 12 subdomains of behavioral functioning: delirium, hallucinations, agitation/aggression, dysphoria, anxiety, euphoria, apathy, disinhibition, irritability/lability and aberrant motor activity, nocturnal behavior disorders, and appetite and eating abnormalities. The NPI is administered to caregivers of dementia patients and a filter question is asked on each subdomain. If the answers to these questions indicate that the patient has problems with a particular subdomain of behavior then the caregiver is asked all the questions about that domain, rating the frequency of symptoms on a 4-point scale, their severity on a 3-point scale, and the distress the symptom causes them on a 5-point scale.

### 2.9. Methods of Data Collection

The information on all the variables foreseen in the study protocol approved by the Ethics Committee was grouped in a computerized data collection form. The database, owned by the INRCA, was built on the INRCA server platform through the establishment of a secure HTTPS protocol. The patient data are kept for the time required by current legislation and in compliance with the protection standards.

### 2.10. Statistical Analysis

Data entry will be performed, providing data entry blocks and checks. Cronbach’s Alpha and other specific tests will be used to assess the quality of the data and their internal consistency. The first questionnaires will be checked manually to assess the completeness of the compilation and any obvious inconsistencies. Next, automated routines will be used to detect outliers and doubtful records. In such cases, necessary data cleaning will be performed. To verify possible distortions due to missed responses during the survey, the characteristics of the subjects in the sample will be compared with those of the non-respondents. The first step of the analysis will be exploratory in nature. The descriptive analysis of the sample will be conducted through the classical techniques of univariate and bivariate statistical analysis. Significant differences between the outcomes and exposures will be compared using the Chi-square test, Fisher’s exact test (categorical variables), the t-test, the ANOVA test, or the corresponding non-parametric U test of Mann-Whitney and the Kruskal Wallis test (comparisons of continuous variables between groups depending on the normal distribution or less of the same. The generalized linear model for repeated measurements will be used to compare the different detection times in addition to the ANOVA model for repeated measurements to evaluate intra-class (before-after) and inter-class differences (case-control). In other words, we will compare the variables that measure the quality of life, depression, and other variables (studied for AD-P) before and after the MBI and also compare the cases with the controls for AD-P. The level of *p* < 0.05 is the significance level.

## 3. Results

The demographis of treated and untreated groups are described in Table 1.

### 3.1. AD Patients Treated with Mindfulness (T0–T1 Six Months)

From the results emerged an improvement in the following dimensions in the AD patients treated with mindfulness after six months:Performance Status Scale ECOG improved (*p* = 0.026);Quality of life SF-36 Tot. improved (*p* < 0.001); An improvement in the following sub-dimensions of SF-36: SF-Physical functioning (PF) (*p* = 0.008); SF-Health role limitation (HR) (*p* < 0.001); SF-Role limitation emotional problems (RE) (*p* < 0.001); SF-Energy-Fatigue-(E) (*p* < 0.001); SF-Emotional well-being (EWB) (*p* < 0.001); SF-Social functioning (SF) (*p* < 0.001); SF-Pain (P) (*p* = 0.009); and SF-General health (GH) (*p* < 0.001) (Table 2).An improvement in spiritual well-being emerged (*p* < 0.001);The depression decreased-BDI (*p* < 0.001);The MMSE remained unchanged (Table 2);The treated patients showed a reduction in the following neuropsychiatric symptoms after six months of treatment: agitation/aggression (*p* = 0.049), anxiety (*p* = 0.002), elation/euphoria (*p* = 0.023), apathy/indifference (*p* < 0.005), irritability (*p* = 0.002), sleep and night time behavior disorders (*p* < 0.005), and appetite/eating disorders (*p* = 0.011) (Table 2).

### 3.2. Untreated AD Patients (T0–T1 Six Months)

The AD Patients without MBI treatment showed significant differences in the following variables after six months:A worsening of MMSE—Mini-Mental State Examination (*p* < 0.001);Performance Status Scale ECOG (*p* < 0.005). The performance status became worse.A worsening in spiritual well-being (SWB) (*p* = 0.021);The depression BDI (*p* < 0.001) became worse;

The Qol and sub-dimensions SF-Emotional well-being (EWB) (*p* = 0.039), SF-Social functioning (SF) (*p* = 0.043), and SF-General health (GH) (*p* = 0.036) became worse (Table 3).

Moreover, the neuropsychiatric symptoms became worse in the untreated patients. In particular, apathy/indifference (*p* = 0.006), disinhibition (*p* = 0.033), and appetite and eating disorders (*p* = 0.013) became worse (Table 4).

No changes emerged in the following dimensions of QoL: SF-Health role limitation, SF-Role limitation emotional problems, and SF-Energy-Fatigue (Table 5).

## 4. Discussion

The findings of this preliminary pilot study show that it is possible to implement a mindfulness-based intervention program in older adults with AD. It can also constitute a low-cost economic intervention.

Our study suggests that mindfulness practice may be a useful non-drug treatment to prevent the worsening of cognitive impairment and restore and maintain a good mood in patients with early-stage Alzheimer’s dementia.

Our results demonstrate that the quality of life and mood improved in these patients after six months of MBI training. Furthermore, the patients trained with MBIs showed an increase in everyday cognition, spiritual well-being, and quality of life and a decrease in depression and neuropsychiatry symptoms. The MMSE cognitive status remained unchanged. The following dimensions of quality of life improved: physical functioning, health role limitation, role limitation emotional problems, energy-fatigue, emotional well-being, social functioning, and pain.

On the other hand, the untreated patients showed a worsening in all the following dimensions after six months: everyday cognition, spiritual well-being, depression, and quality of life. The range of the MMSE became worse. So, the MMSE dimensions orientation, concentration, attention, verbal memory, naming, and visuospatial skills became worse in the untreated patients but remained unchanged in the MBI group after six months of treatment.

This result suggests that an MBI practice can stop the degeneration of some cognitive abilities.

Moreover, the neuropsychiatric symptoms worsened in the non-treated patients. In particular, apathy/indifference, disinhibition, and appetite and eating disorders became worse.

On other hand, the treated AD patients showed a reduction in agitation/aggression, anxiety, elation/euphoria, apathy/indifference, irritability, sleep and nighttime behavior disorders, and appetite/eating disorders.

Furthermore, some studies have shown that apathy is one of the most common neuropsychiatric symptoms (NPS) seen in dementia and is characterized as a multidimensional syndrome of lack of motivation, reduced initiative, akinesia, and emotional indifference [7,8,9,10]. The consequences of apathy are functional impairment, a reduced response to treatment, and increased mortality. Moreover, it is directly correlated with depression, with which some symptoms coincide but are not fully superimposable [43,44], as emerged from a data-driven examination of apathy and depressive symptoms in dementia with independent replication.

Now the decrease in depression in our sample can be an index for the decrease in apathy.

In a study of older adults with subjective cognitive decline, the authors affirmed that there was weaker evidence of an improvement in attention but no effects on executive function [45].

On the other hand, in our study, one of the explanations of the effectiveness of MBI training for the emotional regulation of attention may be the effect of reducing mental distractions by focusing on the mental and emotional contents of the present moment with a non-judgmental attitude. Furthermore, MBI emotional regulation involves a greater ability to maintain a state of attention (alertness) to the activity and improved performance in executive control and memory enhancement tasks, as both behavioral and neuroimaging studies have determined [18,20,21,22,23]. This emotional regulation is effective in reducing stress and problematic psycho-emotional states (e.g., sleep disorders, stress, depression) [28], known as risk factors for the onset and worsening of cognitive performance, mental health, and well-being of the aging population and are associated with an increased risk and worsening of Alzheimer’s disease.

In fact, these psycho-affective states influence well-being in the aging population and are associated with an increased risk of Alzheimer’s disease (AD). Thus, our results showing an improvement in cognitive performance and quality of life and overcoming of the depressive state in our patients can be explained by the effect of reducing anxiety and depression as cognitive/emotional rebalancing as a consequence of the practice of MBI.

The other important aspect of our study is group work. The Alzheimer’s patient at an early stage maintains a lucidity which at times makes him/her aware of all the problems that are occurring, such as those regarding memory and attention. A comparison in a group setting creates conditions for which patients can reduce their experience of discomfort due to their condition and find understanding and support in sharing with others. We must consider that Alzheimer’s disease leads to isolation and creates a sense of unease in relationships. The MBIs group allows the patient to face, integrate, and overcome the emotional stress due to his/her problematic symptoms and, therefore, can be a valid help in dealing with this discomfort.

Based on these considerations, it can be hypothesized that the practice of mindfulness in a group setting could constitute a secondary prevention intervention for Alzheimer’s dementia and other forms of dementia, as well as favoring successful and healthy aging.

We believe that two factors played a fundamental role in determining our results: (1) the emotional rebalancing brought about by awareness (mindfulness practice) and (2) work in a group. Socialization in a specific setting of sharing and solidarity is, in our opinion, fundamental for the emotional well-being of the elderly in general and those who are facing a disease such as Alzheimer’s.

Furthermore, MBI interventions at the onset of Alzheimer’s disease are recommended.

On other hand, we also suggest that the poor effectiveness of mere cognitive training, with little transfer effects in daily life, as reported in the existing scientific literature, could be due to the fact that it does not address the emotional dimension and, thus, does not lead to emotional-affective rebalancing [30,31].

Additionally, most mindfulness application studies involve two-month treatments. In the present study, the patients followed MBI training for six months. We believed that only a longer mindfulness practice (six months or more) could have a positive effect on patients with Alzheimer’s dementia given the complexity of the pathology, which involves the degeneration of the entire personality. This may be an indication for further studies in the future.

Furthermore, it is hoped that future studies will compare the effectiveness of mindfulness practices applied at different times in patients with early-stage Alzheimer’s disease (e.g., three months, six months, or nine months). Furthermore, we hypothesize that mindfulness training must, in these patients, be continued and/or repeated over time to maintain efficacy (e.g., two to three months of practice after a year of the longest training).

Additionally, future studies could apply MBIs to patients with other levels of Alzheimer’s disease severity to test its effectiveness, even when the disease has resulted in more personality degeneration.

We also hypothesize that given the Medicare costs for Alzheimer’s disease (we must consider not only the patients but also the psycho-physical health of the family members), an early secondary prevention intervention could reduce them. Further studies could evaluate the economic impact of intervention with MBIs on AD patients and their caregivers.

The first limitation of the present study is the small sample size, and care is needed when extrapolating these results to other stages of the disease. Second, a sampling bias was present in the data because all the subjects attended only one institution and, thus, were not representative of AD patients in general. Another limitation of the study is the lack of comparison between females and males. Such a comparison was not possible due to the limited number of cases. Hopefully, further studies will investigate these issues.

## 5. Conclusions

This study can provide insight into mindfulness intervention, which may be useful in preventing degeneration in early-stage AD patients and restoring and maintaining a good quality of life. For these reasons, our results are of clinical and practical importance. We suggest that MBIs can be especially effective at the onset of Alzheimer’s disease.

## Figures and Tables

**Table 1 brainsci-13-00484-t001:** Patients-Demographic characteristics (treatment and no treatment groups).

	Mindfulness (n = 22)	No Treatment (n = 22)	*p*
Age (years), mean ± sd	82.8 ± 5.6	82.9 ± 4.2	0.855
Gender (female), n (%)	14 (63.6)	17 (77.3)	0.322
Marital status, n (%)			0.157
*Single*	13 (59.1)	10 (45.5)	
*Married*	7 (31.8)	12 (54.5)	
*Divorced*	2 (9.1)	0 (0.0)	
Living with, n (%)			0.127
*Alone*	4 (20.0)	8 (36.4)	
*With spouse*	10 (50.0)	10 (45.5)	
*With spouse and unmarried children*	1 (5.0)	0 (0.0)	
*With unmarried children*	1 (5.0)	2 (9.1)	
*With married children*	0 (0.0)	2 (9.1)	
*Other*	4 (20.0)	0 (0.0)	
Number of sons, mean ± sd	2.4 ± 1.0	1.9 ± 0.6	0.071
Educational level, n (%)			0.223
*Elementary School*	12 (57.1)	16 (72.7)	
*Middle School*	2 (9.5)	4 (18.2)	
*High School*	3 (14.3)	2 (9.1)	
*University*	4 (19.0)	0 (0.0)	

**Table 2 brainsci-13-00484-t002:** Mindfulness-Patients: Evaluation of quality of life, depression, and MMSE (T0–T1 six months).

	Test	Retest	*p*
MMSE	21.4	22.3	0.286
ECOG—THE EVERYDAY COGNITION	1.8	1.3	0.026
SWB—SPIRITUAL WELL-BEING	21.8	27.6	<0.001
BDI—DEPRESSION	16.3	2.2	<0.001
SF—HYSICAL FUNCTIONING	46.1	64.3	0.008
SF—HEALTH ROLE LIMITATION	14.8	92.0	<0.001
SF—ROLE LIMITATION EMOTIONAL PROBLEM	22.7	98.5	<0.001
SF—ENERGY-FATIGUE	46.4	67.7	<0.001
SF—EMOTIONAL WELL-BEING	55.1	79.6	<0.001
SF—SOCIAL WELL-BEING	55.1	97.7	<0.001
SF—PAIN	59.2	80.0	0.009
SF—GENERAL HEALTH	51.1	70.5	<0.001

**Table 3 brainsci-13-00484-t003:** Untreated patients: Evaluation of quality of life, depression, and MMSE (T0–T1 six months).

	Test	Retest	*p*
MMSE—MINI-MENTAL STATE	21.3	17.3	<0.001
ECOG—THE EVERYDAY COGNITION	2.0	2.8	0.005
SWB—SPIRITUAL WELL-BEING	20.8	18.2	0.021
BDI-DEPRESSION	16.2	26.5	<0.001
SF—PHYSICAL FUNCTIONING	39.5	29.3	0.197
SF—HEALTH ROLE LIMITATION	15.9	3.4	0.119
SF—ROLE LIMITATION EMOTIONAL PROBLEM	27.3	13.6	0.214
SF—ENERGY-FATIGUE	43.9	33.0	0.064
SF—EMOTIONAL WELL-BEING	54.2	42.9	0.039
SF—SOCIAL WELL-BEING	53.4	37.5	0.043
SF—PAIN	62.5	46.7	0.064
SF—GENERAL HEALTH	50.9	39.3	0.036

**Table 4 brainsci-13-00484-t004:** Untreated patients: Evaluation of neuropsychiatric symptoms (T0–T1 six months).

NPI Frequency Score			
Delirium	0.8	1.2	0.351
Hallucinations	0.4	0.8	0.249
Agitation—Aggression	1.8	2.4	0.058
Depression	1.8	2.3	0.259
Anxiety	2.0	2.7	0.128
Elation—Euphoria	0.5	0.5	0.735
Apathy—Indifference	1.8	3.0	0.006
Disinhibition	0.2	0.8	0.033
Irritability	2.5	2.6	0.629
Aberrant motor behavior	0.9	1.7	0.056
Sleep and nighttime behavior disorders	1.9	2.5	0.202
Appetite and eating disorders	0.8	2.0	0.013

**Table 5 brainsci-13-00484-t005:** Mindfulness- Patients: Evaluation of neuropsychiatric symptoms (T0–T1 six months).

NPI Frequency Score			
Delirium	0.5	0.2	0.226
Hallucinations	0.4	0.1	0.185
Agitation—Aggression	1.1	0.5	0.049
Depression	1.0	0.5	0.059
Anxiety	1.9	0.9	0.002
Elation—Euphoria	0.7	0.1	0.023
Apathy—Indifference	1.5	0.6	0.005
Disinhibition	0.4	0.2	0.351
Irritability	1.6	0.6	0.002
Aberrant motor behavior	0.5	0.1	0.073
Sleep and nighttime behavior disorders	0.9	0.1	0.005
Appetite and eating disorders	1.0	0.2	0.011

## Data Availability

The data can be provided under justified requests to the authors (Maria Velia Giulietti and Anna Vespa).

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
