# Peer review of "Effects of Mindfulness-Based Interventions (MBIs) in Patients with Early-Stage Alzheimer’s Disease: A Pilot Study"

_brainsci, 2023, doi:10.3390/brainsci13030484_

Round 1

Reviewer 1 Report

This pilot study show some effects of mindfulness based interventions (MBIs) in a small group of patients with early stage Alzheimer’s disease. Most clinical symptoms the authors checked was significantly changed in the treatment groups. Although the data is limited due to some reasons the authors already mentioned in the discussion, it is still an interesting study. 

Author Response

I thank the reviewer on behalf of all authors for his/her favorable opinion on the article.

Reviewer 2 Report

Very interesting with remarkable results.

- regression and/or correlation analyses may provide an element of predictiveness.

- associations among the assessed variables would contribute to the Discussion which is somewhat meagre.

- although they mention 'limitations' a subsection would be better.

Author Response

I thank the reviewer on behalf of all authors for his/her favorable opinion on the article and for the suggestions.

The English was corrected.

Reviewer 3 Report

This is a well-structured article. The main question addressed by this research is the effects of Mindfulness Based Interventions (MBIs) in patients with early stage AD.

The introduction gives the background of this study as it briefly describes the current knowledge about AD as a neurodegenerative disease, its main therapeutic interventions and the potential role of MBIs.

“Materials and Methods” section is descriptive enough. It refers to the subjects of the study, the inclusion and exclusion criteria used, the randomization process, the measures and the statistical analyses that were implemented during this study. One methodological issue that, to my opinion, is important is the fact that the control group was left without any intervention.

The results are quite interesting and, to my opinion, well presented.

The discussion is well written, summarizing and discussing the main findings of the study. The existence of a paragraph summarizing its main limitations is, to my opinion, one advantage of this study too.

Furthermore, the conclusions could be written in a more detailed manner, perhaps proposing some specific targets for future studies.

References, although relatively few, are relative to the subject.

English language and style are generally fine. Nevertheless, there are several issues that need to be addressed before publication. For example, after the words “furthermore”, “moreover”, “thus” a “,” should be added. In the discussion section, the term “one institutions” should be replaced by “one institution” and in the conclusion section the term “these reason” should be replaced by “these reasons”.

Author Response

I thank the Reviewer on behalf of all authors for his/her favorable opinion on the article and for the suggestions.

Based on the suggestions, we revised the manuscript in its parts (introduction, methodology, results, etc.) and added reflections on possible future studies in the discussion.

The explanation of the fact that the control group was left without any intervention was added in the methodology.

All additional parts have been highlighted in yellow (n.words 4019)

Round 2

Reviewer 3 Report

To my opinion, the manuscript has been revised well.